# Percutaneous Anorectoplasty (PARP)—An Adaptable, Minimal-Invasive Technique for Anorectal Malformation Repair

**DOI:** 10.3390/children9050587

**Published:** 2022-04-21

**Authors:** Julia Küppers, Viviane van Eckert, Nadine R. Muensterer, Anne-Sophie Holler, Stephan Rohleder, Takafumi Kawano, Jan Gödeke, Oliver J. Muensterer

**Affiliations:** 1Department of Pediatric Surgery, Dr. von Hauner Children’s Hospital, Ludwig-Maximilians-University Medical Center, 80337 Munich, Germany; julia.kueppers@med.uni-muenchen.de (J.K.); viviane.van@med.uni-muenchen.de (V.v.E.); nadinemuensterer@icloud.com (N.R.M.); annesophie.holler@med.uni-muenchen.de (A.-S.H.); jan.goedeke@med.uni-muenchen.de (J.G.); 2Department of Pediatric Surgery, Johannes-Gutenberg-University Medical Center Mainz, 55131 Mainz, Germany; stephan.rohleder@unimedizin-mainz.de; 3Department of Pediatric Surgery, Kagoshima University, Kagoshima 890-8520, Japan; kawano-t@m2.kufm.kagoshima-u.ac.jp

**Keywords:** anorectal malformation, percutaneous, anorectoplasty, ultrasound, fluoroscopy, endoscopy, keyword, perineal fistula, Down syndrome

## Abstract

Background: Anorectal malformations comprise a broad spectrum of disease. We developed a percutaneous anorectoplasty (PARP) technique as a minimal-invasive option for repair of amenable types of lesions. Methods: Patients who underwent PARP at five institutions from 2008 through 2021 were retrospectively analyzed. Demographic information, details of the operative procedure, and perioperative complications and outcomes were collected. Results: A total of 10 patients underwent the PARP procedure during the study interval. Patients either had low perineal malformations or no appreciable fistula. Most procedures were guided by ultrasound, fluoroscopy, or endoscopy. Median age at PARP was 3 days (range 1 to 311) days; eight patients were male. Only one intraoperative complication occurred, prompting conversion to posterior sagittal anorectoplasty. Functional outcomes in most children were highly satisfactory in terms of continence and functionality. Conclusions: The PARP technique is an excellent minimal-invasive alternative for boys born with perineal fistulae, as well as patients of both sexes without fistulae. The optimal type of guidance (ultrasound, fluoroscopy, or endoscopy) depends on the anatomy of the lesion and the presence of a colostomy at the time of repair.

## 1. Introduction

Anorectal malformation affects around 1:5000 liveborn infants and comprises a wide spectrum of conditions concerning the distal anus, rectum, as well as urogenital tract. Half of all anorectal malformations are considered low anorectal malformations [1,2,3].

In 1982, Peña and de Vries introduced the posterior sagittal anorectoplasty (PSARP), which has become the standard of open repair [1,4]. The procedure, however, is associated with a relatively high risk of dehiscence, wound infection, stricture, and long-term continence issues [3,5,6]. Some forms of anorectal malformations may be amenable to a less invasive repair in which the neorectum is reconstructed through the intact sphincter. For high lesions, including rectoprostatic and rectovestibular fistulas, the laparoscopy-assisted anorectoplasty (LAARP) procedure, first introduced by Georgeson et al. in 2000, has become routine in many centers around the world [7]. Lower lesions, however, such as rectoperineal fistulas in boys, and imperforate anus without a fistula, do not require or benefit from laparoscopy [8].

We therefore devised a minimal-invasive percutaneous procedure that avoids opening of the pelvic floor or splitting the sphincter, the so-called PARP. Over the course of more than a decade, this procedure has undergone several important modifications. The aim of this study is to retrospectively describe and evaluate all patients who underwent a PARP procedure in terms of feasibility, effectiveness, complications, and outcome.

## 2. Materials and Methods

### 2.1. Study Design

This is a retrospective analysis of all patients who underwent the PARP procedure at 5 different institutions from 2008 through 2021. In order to be eligible for inclusion, the patients had to be either male with a typical perineal fistula or of either sex without a fistula. All other forms of anorectal malformation were excluded.

### 2.2. Ethics

The study was approved by the ethics board of the Ludwig Maximilian University Faculty of Medicine (registration number 22-0141). The parents or caregivers gave their explicit and written consent on having their child operated on using this novel method. Alternatives were described and offered. The potential risks and benefits were discussed in detail.

### 2.3. Operative Technique

#### 2.3.1. PARP without Image Guidance (nPARP)

The percutaneous anorectoplasty (PARP, Figure 1, Appendix A: Description of the PARP procedure without image guidance in a newborn male with anorectal malformation and a rectoperineal fistula) is applicable in patients born with pure imperforated anus covered by a skin tag in the form of a bucket handle. A Wangensteen–Rice invertogram radiograph confirms the presence of a low imperforated anus. The bucket handle is dissected off the underlying tissue in a modified lithotomy position. Subsequently, the bucket handle is divided and resected. The posterior skin is excised in a wedge-type fashion. Then, the muscle complex is located at the center of the neo-anus using an electronic muscle stimulator. Once the sphincter complex has been detected, a needle or an eight French dilator is passed through the sphincter complex into the rectal pouch. A guidewire is passed through the rectal pouch to secure the tract. Once secured, the tract is dilated sequentially from eight to twenty French. At that point, the guidewire and dilator are removed. Following the evacuation of meconium, Hegar dilators are used to dilate the tract further to approximately eleven millimeters or an age-appropriate size. Finally, retractors are inserted. The rectal mucosa is retracted down to the level of the skin and is then sewn to the skin using interrupted polyglactin sutures resulting in a cosmetically pleasing, inverted, and orthotopic anus with an intact sphincter complex.

#### 2.3.2. Ultrasound-Guided PARP (uPARP)

When performing an ultrasound-guided percutaneous anorectoplasty (uPARP), the previously described PARP operative technique is complemented by real-time ultrasound imaging of the perineum in the operating room to localize the rectum and muscle complex (Figure 2).

#### 2.3.3. Fluoroscopy-Guided (Interventional) PARP (iPARP)

During a fluoroscopy-guided (interventional) percutaneous anorectoplasty (iPARP), the patient is placed in a prone position with the buttocks elevated, much like during a conventional posterior sagittal anorectoplasty (PSARP). The fluoroscopy unit is positioned in a cross-table lateral configuration. The center of the muscle complex is identified using an electronic stimulator. A needle is advanced through the center of the sphincter into the air-filled rectal pouch under fluoroscopic guidance and the guidewire is advanced through the needle (Figure 3a,b). A 12 mm balloon dilator is advanced over the needle and the tract is dilated (Figure 3c,d). Thereafter, the mucosa is retracted down to the skin using hooks and sutured circumferentially as described for the PARP above.

#### 2.3.4. Endoscopically-Guided PARP (ePARP)

An endoscopically-guided percutaneous anorectoplasty (ePARP, Figure 4, Appendix A: Description of the ePARP procedure in a 6 month old girl with Down syndrome who had a transverse colostomy in an outside hospital) requires a previous colostomy and is thus performed in children without a fistula, usually patients with Down syndrome. The patient is placed supine in a way that allows a distal colonoscopy from the mucous fistula. A fluoroscopy unit is placed to allow cross-table lateral imaging. At the blind end of the colon, a typical star-shaped scar is always detected and marks the center of the future tract (Figure 5). The center of the muscle complex is identified from the outside with a stimulator and a needle is advanced through the sphincter complex into the rectum under X-ray and endoscopic guidance. A guidewire is placed. Then, a twelve-millimeter balloon dilator is inserted over the guidewire and inflated to dilate the tract. After the balloon is deflated, the tissue tract can be inspected endoscopically to the outside. Subsequently, the endoscope is retracted back inside. The next step involves bringing the rectal mucosa down to the skin. This is accomplished by introducing two sharp hooks, one anteriorly and one posteriorly, which gently retract the mucosa. From the outside, circular stay sutures are placed on the mucosal sleeve. The exact placement of the sutures can be verified endoscopically. Thereafter, a colocutaneous anastomosis is performed using circular braided absorbable sutures. Correct placement of the sutures can be verified endoscopically to confirm that the mucosa circularly anastomoses with the skin. This is important to prevent stricture. Finally, the stay sutures are cut leaving a watertight anastomosis. At the end of the procedure, the neo-anus is calibrated using a ten-millimeter Hegar dilator.

### 2.4. Data Acquisition

The data were retrospectively collected from operative reports and hospital records. Pertinent demographic information, comorbidities, operative time, type of PARP, perioperative and postoperative complications, as well as short- and long-term outcome were extracted into a database.

## 3. Results

### 3.1. Patients

During the study interval, a total of 10 patients were included. Eight of those patients were male. Half of the patients presented with anorectal malformation with a perineal fistula; the other half did not exhibit an appreciable fistula. Only three patients did not show comorbidities. Three patients were diagnosed with Down syndrome, one patient suffered from VACTERL, and one patient presented with Currarino triad, Spina bifida, as well as congenital heart disease. Furthermore, two patients were born prematurely, one of whom experienced a pneumothorax and underwent chest tube placement preoperatively. This patient was later diagnosed with Duchenne muscular distrophy. Moreover, four patients had received a colostomy prior to the PARP procedure. The median age for colostomy placement was 1.5 days (range 0 to 2) (Table 1).

### 3.2. Operations

Overall, ten percutaneous anorectoplasties were performed between 2008 and 2021. The median age at the PARP was three days (range 1 to 311 days). The median operative time amounts to approximately 60 min (range 25 to 183). The OP times of the final two PARP procedures were not included in this calculation as patients underwent multiple concomitant surgical procedures. Apart from the initial two percutaneous anorectoplasties without image guidance, the procedures were generally guided: one uPARP, three iPARPs, and four ePARPs were performed (Table 2).

### 3.3. Complications

There was one complication in the second child who was operated on without image guidance. The procedure was initiated in the supine position with the legs raised and hips flexed. Preoperatively, a Foley bladder catheter was placed, but it could not be advanced all the way and no urine was obtained. It was left in place without inflating the balloon. After punctuation of the rectum and dilation, the Foley catheter was visible through the rectum, prompting us to abort the procedure. The patient was turned prone, prepped, and draped. Then, a posterior sagittal anorectoplasty (PSARP) was performed. Subsequently, the Foley catheter was removed, a new catheter was placed through the urethra under vision into the bladder, and the bulbar urethral opening, where the first catheter had passed into the rectum, was repaired using interrupted resorbable sutures. No other peri- or postoperative complications were noted in this series (Table 2). There were no wound infections.

### 3.4. Outcomes

The median follow-up lasted approximately 16 months (range 0 to 43. Table 3). Two out of ten patients dealt with constipation postoperatively, one of which required oral macrogol (polyethylene-glycol) treatment. None of the patients suffered from incontinence following the PARP procedure. Four patients required further dilations. Overall, outcomes were highly satisfactory in most patients in terms of functionality and continence.

## 4. Discussion

This is the largest case series on percutaneous anorectoplasty to date. Over the course of the last decade, the technique has undergone evolution using additional image guidance, to the point where it can be safely performed and recommended for certain anorectal malformations.

Despite the heterogeneous pattern of anorectal malformations (ARMs) [1], posterior sagittal anorectoplasty (PSARP) has been the main approach for repair across the board. The drawback of PSARP is the division of the sphincter in two halves through the midline, with later reconstruction [1,9]. Despite the argument that this allows accurate visualization of anatomical structures, thus allowing the most accurate surgical correction and preservation of blood vessels and nerve structures, current studies in the literature increasingly consider that the invasiveness of this method may not be necessary in certain cases [2,10,11]. Laparoscopy can assist with repair of high forms of ARMs while leaving the sphincter intact [7,10], although the intuitive hypothesis that this could improve the functional prognosis in terms of decreasing cases of fecal incontinence and constipation has not yet been conclusively confirmed [9,10,12,13]. Nevertheless, the laparoscopic, sphincter-sparing approach indeed has been shown to significantly reduce postoperative wound complications and hospital stay [9,10,12,14]. Since wound complications have a negative impact on functional prognosis, the advantage of minimally invasive techniques is increasingly evident [1,3,4,9,10,12,14,15]. Other approaches to reduce wound complications such as preoperative bowel preparation, prolonged postoperative fasting and antibiotics, as well as application of a vacuum-assisted pump have also been described with varying degrees of success [4,16,17,18]. While laparoscopy is useful for high forms of anorectal malformations, it is not as helpful for low lesions.

To date, there are only a few reports describing minimal invasive techniques for low lesions. Pakarinen et at. described the “Transanal Endoscopic-Assisted Proctoplasty (TEAPP)” [2,11]. They performed a sigmoidostomy in seven patients with ARM without a fistula in term of a staged surgical approach. Via colostomy, the absence of a fistula was confirmed (high-pressure colostogram) before implementing the TEAPP procedure. A retrograde endoscopy through the sigmoid mucous fistula was performed to visualize the termination of the rectum. In case a low malformation was confirmed by using translumination of the endoscope light from the rectum to the anal dimple within the external sphincter, correction via TEAPP was performed (successful in four of the seven patients). The rectum was incised from below and the neoanus was created under endoscopic visual control, similar to the ePARP procedure described in our report. They suggested that this technique allows anatomical reconstruction of the anorectum, by placing the anorectum within the sphincter complex under endoscopic control [11]. In this study, the TEAPP procedure was aborted and converted to a PSARP in three of the seven recruited patients, mainly because transillumination could not be positively confirmed. The question of transillumination raises the question of the maximal distance between skin and pouch in those without a fistula that is repairable by ePARP. In our series, the maximal distance was 3 cm. Using the hooks, it was still feasible to bring the mucosa down to the anus without difficulties for anastomosis. Nevertheless, the distance between the pouch and the skin may be a limitation of the PARP technique, making it applicable only to low-type lesions where the mucosa can be retracted downward and anastomosed to the skin. This approximation, however, results in a nicely inverted skin rosette and may prevent prolapse, which we have not seen as a complication in our series.

While another option may be to perform a limited perineal skin incision to access the distal rectal pouch under direct vision, we believe that using ultrasound, radiography, or endoscopy allows us to penetrate through the center of the sphincter complex with a needle, limiting dissection and associated damage, much like during the laparoscopic approach for higher lesions.

In contrast to the generally accepted concept that even in low forms of ARMs without fistula there is intimate contact between the rectal blind sac and the posterior urethra [1], Pakarinen et al. describe the midpoint of the distal rectal termination to be right above the anal site within the sphincter muscle complex and not intimately related to the urethra. This finding may disprove the argument that the close relationship between the rectum and the urethra justifies the need for PSARP in low forms of ARMs [2,11]. 

The results regarding the percutaneous anorectoplasty procedure (PARP) described in this article show comparable advantages to the TEAPP procedure. The minimally invasive approach may help avoid potential complications associated with PSARP in select, eligible patients. Furthermore, the high success rate in our study (90%; only one patient was converted to a PSAPR procedure) indicates that suitable cases can be reliably identified preoperatively.

In contrast to TEAPP, the PARP procedure allows, in addition to minimal invasive correction of patients with low ARMs, the correction of male patients with a perineal fistula (anocutaneous, rectoperineal outside the sphincter complex). These types of malformations are currently still recommended to be reconstructed by posterior sagittal anoplasty [19]. However, there is evidence suggesting that overall functional outcome is comparable after minimally invasive anoplasty and PSARP for perineal fistula in boys [20]. Additionally, in contrast to the TEAPP, the ePARP in our series is performed not only using endoscopic guidance, but under concomitant fluoroscopic control. In our opinion, this is essential for a safe, precise reconstruction of the anorectum.

Obviously, the ePARP procedure requires a prior colostomy for antegrade endoscopy, but also for the preoperative exclusion of an occult rectourethral or rectovesical fistula by high-pressure distal colostogram [19]. However, the iPARP procedure does not require a colostomy and therefore may be an option when the invertogram clearly shows the blind-ending rectum and there are no signs of a fistula.

Relevant complications of colostomies in newborns include wound complications, prolapse, leakages, parastomal hernias, or bowel obstruction [21]. Therefore, colostomies should be avoided if possible, particularly in males with perineal fistulas. Another argument in favor of a one-stage procedure is the so-called “brain–defecation reflex” that may remain intact following the “use it or lose it” principle [22,23]. Finally, there is evidence of one-stage procedures affording similar outcomes compared to multi-stage procedures. This raises the question whether liberal placement of a colostomy is generally warranted [9,16,24].

In our series, only one perioperative complication occurred during the PARP procedure, namely, the presence of the Foley catheter in the rectum upon visualizing the rectum from the perineum. The unanswered question remains whether the posterior urethra was injured during the procedure or whether the patient had a low rectourethral fistula in addition to the perineal fistula in the first place. According to the literature, such H-type anorectal malformations have an incidence of around 3 percent [25], ranging from 0.1 to 16 percent [26]. Therefore, pediatric surgeons should have a high index of suspicion when performing any of these procedures. Conversion to a PSARP in our case 2 afforded the patient a good outcome. Surgeons attempting a PARP procedure should maintain a high index of suspicion for rectourethral fistulae and should convert to PSARP if there is any indication that anatomy is not as preoperatively suspected. In our case, the patient did not have a micturating cysturethrogram, which would have been helpful. To ensure patient safety, accurate preoperative evaluation of the underlying anatomy and, accordingly, the selection of the appropriate surgical technique is crucial. This refers to the level of the ARM, the relation of the rectal pouch to the muscle complex as well as the evaluation of a rectogenitourinary communication [1,2,19]. These aspects may be estimated by a lateral pelvic radiograph, ultrasound, cystoscopy, or micturating cystourethrogram (MCUG), even though the results of these examinations may be inaccurate in some cases [2,10,27].

There were no complications during the ePARP procedures throughout our study. In our opinion, the ePARP procedure, including employing intraoperative fluoroscopy, offers the safest technique, especially in cases where preoperative diagnostics have not provided complete clarity regarding the exact type of ARM. The relevance of accurate preoperative diagnostics also applies to perioperative guidance. Using a percutaneous technique without some kind of image guidance (nPARP) has a high potential risk of creating false tracks and causing complications in neighboring structures such as the urethra, as seen with patient number 2 in this series. We therefore do not recommend performing the nPARP procedure.

Functional outcomes in most children were highly satisfactory in terms of continence and functionality, with only two cases of constipation and four patients with the need of anal dilations. We are aware that the follow-up in this study was too short to draw any conclusion concerning long-term functional outcomes. However, there is evidence showing that long-term results in low malformations are good in most patients if perioperative complications are prevented [1,2,19]. Furthermore, long-term follow-up of these patients in terms of functionality remain controversial and is generally influenced by confounding factors, including a high incidence of associated anomalies [20,28].

The study is limited by its relatively small sample size, the retrospective design and heterogeneous population. Furthermore, the technique was implemented by multiple surgeons. However, all surgeons had comparable experience in the field of pediatric surgery and had discussed the exact technique prior to the interventions. These factors mean that our data can only demonstrate a trend and that, so far, no precise statement can be made about certain secondary endpoints, such as the operating room time.

This is the first study investigating the clinical outcome after PARP procedure as well as describing the different options of visual guidance. The PARP procedure seems to offer a safe and individually tailored minimally invasive surgical approach to avoid unnecessary invasive surgery in eligible patients. Prospective studies with larger populations are needed to confirm these findings.

## Figures and Tables

**Figure 1 children-09-00587-f001:**
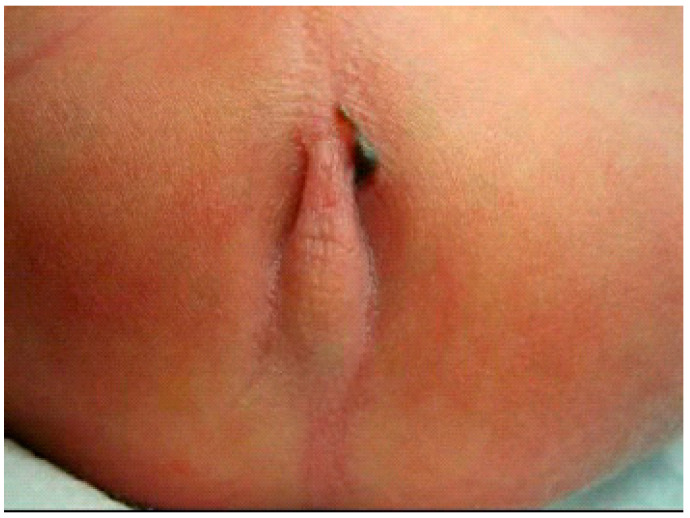
Screenshot of Appendix A. Typical perineal fistula with bucket-handle in a boy.

**Figure 2 children-09-00587-f002:**
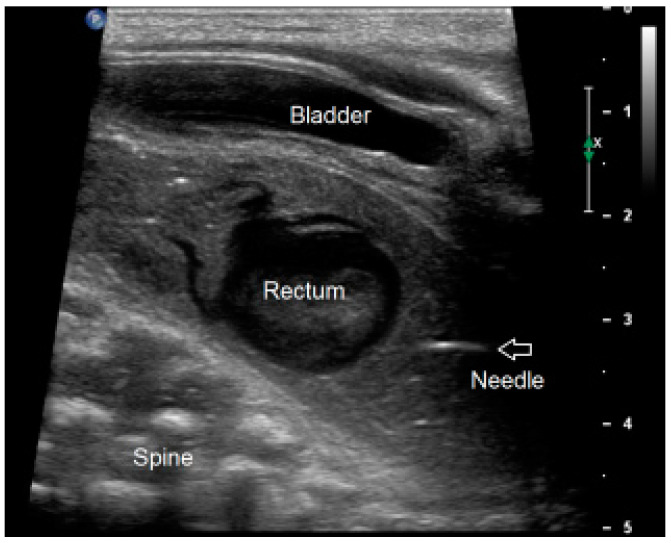
Ultrasound image during uPARP showing the guidance of the needle (arrow) towards the meconium-filled rectal pouch.

**Figure 3 children-09-00587-f003:**
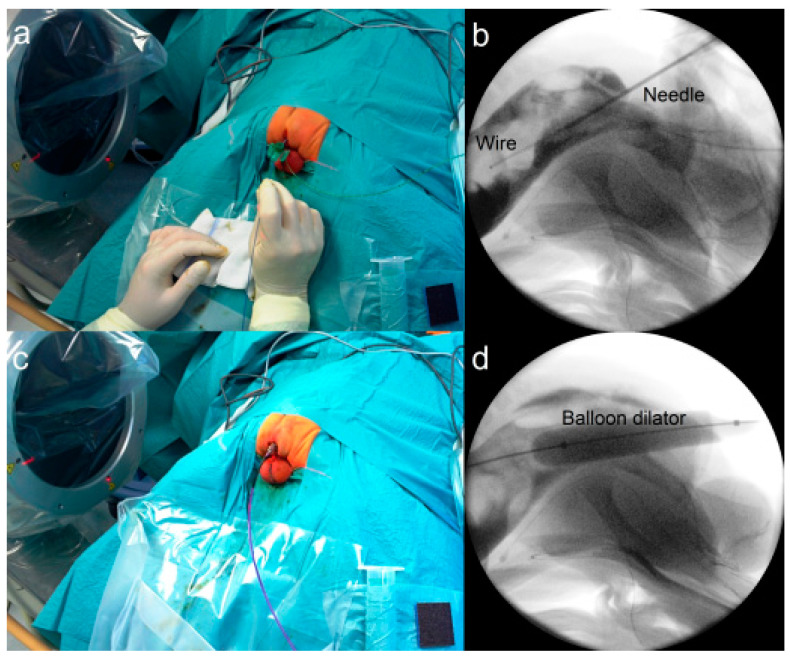
Technique of iPARP. The needle and guidewire are introduced into the rectum through the sphincter complex (**a**) under fluoroscopic guidance (**b**). A balloon dilator is advanced over the guidewire (**c**) to dilate the tract (**d**).

**Figure 4 children-09-00587-f004:**
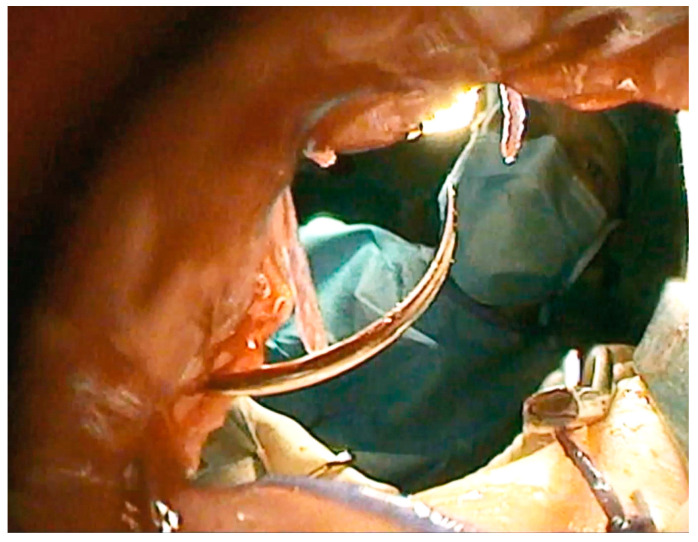
Screenshot of Appendix A. Trans-neoanal endoscopic view of the anastomosis being con-structed.

**Figure 5 children-09-00587-f005:**
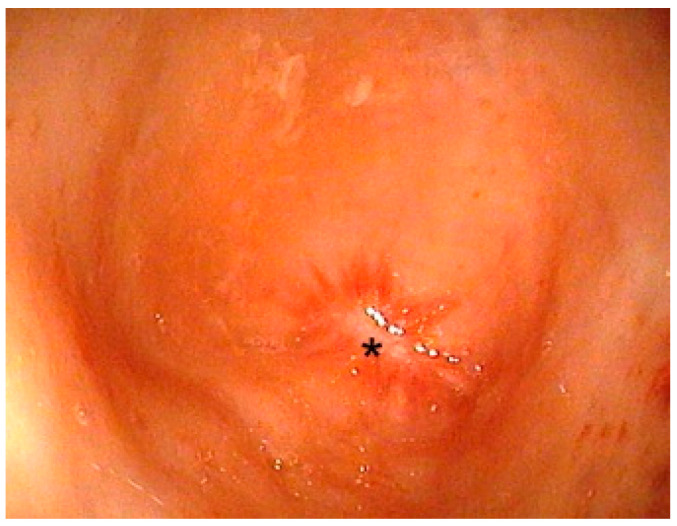
Star-shaped end of the rectal pouch marking the future tract towards the sphincter complex (*).

**Table 1 children-09-00587-t001:** Patient demographics and data (VACTERL—vertebral, anorectal, cardiac, tracheo-esophageal, renal, limb anomalies).

Patient	Sex	Year	Type of Anorectal Malformation	Comorbidities	Colostomy	Age (Days) at Colostomy
1	Male	2008	Perineal fistula, bucket handle	None	No	-
2	Male	2010	Perineal fistula (pinpoint at raphe)	None	No	-
3	Female	2011	No fistula	Down syndrome	No	-
4	Male	2011	Perineal fistula	32-week prematurity, left pneumo-thorax and chest tube placement	No	-
5	Male	2015	Perineal (scrotal) fistula	36-week prematurity, diagnosed later with Duchenne	No	-
6	Male	2015	Perineal fistula	Currarino triad, Spina bifida, congenital heart disease	No	-
7	Female	2015	No fistula	Down syndrom	Yes	3
8	Male	2017	No fistula	None	Yes	3
9	Male	2020	No fistula	Down Syndrome	Yes	1
10	Male	2021	No fistula	VACTERL association	Yes	2

**Table 2 children-09-00587-t002:** Operative data and complications. (* Total operative time includes ePARP and other procedures, namely, cystoscopy, esophagoscopy, Kimura-lengthening of upper esophageal pouch. ^§^ Operative time includes cystoscopy).

Patient	Age at PARP (Days)	Operative Time (Minutes)	Type of PARP	Intraoperative and Perioperative Omplications
1	2	25	No image guidance	None
2	2	183	No image guidance	Injured urethra or second fistula, converted to psarp, urethra repaired, colostomy performed
3	3	68	uPARP	None
4	1	57	iPARP	None
5	1	109	iPARP	None
6	3	44	iPARP	None
7	225	51	ePARP	None
8	77	62	ePARP	None
9	168	236 *	ePARP	None
10	311	111 ^§^	ePARP	None

**Table 3 children-09-00587-t003:** Postoperative data and outcomes (y—years, m—months).

Patient	Age at Last Follow-Up	Constipation	Incontinence	Dilations	Additional Comments
1	2 y 3 m	No	No	No	Potty trained at 2 years, functionally normal
2	2 m	-	-	Yes	Short-term well, long-term lost to follow-up
3	1 y 3 m	Yes	No	Yes	Needed macrogol, otherwise no problems in the follow-up time period
4	1 y 8 m	No	No	No	Started potty training
5	3 y 9 m	No	No	No	No problems, normal stooling pattern, general hypotonia due to duchenne muscular dystrophy in toddlerhood
6	6 m	No	No	Yes	Died of congenital heart disease at 6 months
7	2 y 10 m	No	No	No	Colostomy takedown at 9 months of age, normal spontaneous defecation pattern 1× per day
8	9 m	No	No	No	Colostomy performed at the umbilicus, no issues with stooling, no medications
9	1 y 11 m	No	No	No	Started potty training
10	8 m	Yes	No	Yes	Too early to evaluate continence

## Data Availability

All data on which this publication is based are available from the corresponding author (OM) upon reasonable request.

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
