# Peer review of "Percutaneous Anorectoplasty (PARP)—An Adaptable, Minimal-Invasive Technique for Anorectal Malformation Repair"

_children, 2022, doi:10.3390/children9050587_

Round 1
Reviewer 1 Report
Review Children manuscript 1649367: Percutaneous Anorectoplasty (PARP) - an adaptable, minimal invasive technique for anorectal malformation repair by Küppers et al.
The authors present a retrospective analysis of anorectal malformation (ARM) patients with no or a low fistula undergoing percutaneous anorectoplasty in 5 institutions between 2008 and 2021. These patients typically undergo a posterior sagittal anorectoplasty for repair of their ARM, but the authors adapted their technique from the laparoscopy-assisted anorectoplasty for ARMs with high fistulas. The aim of their study was to summarize their experience with the new technique. The authors present descriptive data on a total of 10 patients. They used PARP without image guidance in 2 patients, ultrasound guided PARP in 1 patient, fluoroscopy guided PARP in 3 patients and endoscopy guided PARP in 4 patients. They provide clear descriptions of the different techniques and good descriptive data on outcomes. The discussion is balanced and highlights strengths and weaknesses of this new technique. Overall, this is a well written manuscript and I support publication in the special issue on Current Development of Pediatric Minimally Invasive Surgery in Children. My only concern is with PARP without some kind of image guidance. This in my mind has a high potential risk of creating false tracks and causing complications in neighboring structures such as the urethra etc. The authors state that they do not recommend the nPARP procedure, but I feel that they can expand this section of the discussion a bit by mentioning the potential problems and complications that can happen when this technique is implemented without image guidance.
Author Response
Responses to the reviewers:
- Reviewer 1
The authors present a retrospective analysis of anorectal malformation (ARM) patients with no or a low fistula undergoing percutaneous anorectoplasty in 5 institutions between 2008 and 2021. These patients typically undergo a posterior sagittal anorectoplasty for repair of their ARM, but the authors adapted their technique from the laparoscopy-assisted anorectoplasty for ARMs with high fistulas. The aim of their study was to summarize their experience with the new technique. The authors present descriptive data on a total of 10 patients. They used PARP without image guidance in 2 patients, ultrasound guided PARP in 1 patient, fluoroscopy guided PARP in 3 patients and endoscopy guided PARP in 4 patients. They provide clear descriptions of the different techniques and good descriptive data on outcomes. The discussion is balanced and highlights strengths and weaknesses of this new technique. Overall, this is a well written manuscript and I support publication in the special issue on Current Development of Pediatric Minimally Invasive Surgery in Children.
1.1. Critique: My only concern is with PARP without some kind of image guidance. This in my mind has a high potential risk of creating false tracks and causing complications in neighboring structures such as the urethra etc. The authors state that they do not recommend the nPARP procedure, but I feel that they can expand this section of the discussion a bit by mentioning the potential problems and complications that can happen when this technique is implemented without image guidance.
> We totally agree, thank you for the comment. We have expanded the respective paragraph on page 8, now reading "Using a percutaneous technique without some kind of image guidance (nPARP) has a high potential risk of creating false tracks and causing complications in neighboring structures such as the urethra, as seen with Patient number 2 in this series. We therefore do not recommend performing the nPARP procedure."
Reviewer 2 Report
The authors present their experience with "percutaneous anorectoplasty" with low-infralevator anorectal malformations. Although the technique is described previously, this case series is presented with a wide range of percutaneous options in a relatively small number of patients which also represents a relatively rarer form of the malformation (as evidenced by only 10 patients in a 13 year period from multi-institutional collaboration).
The text is well written, clear and simple, and easy to follow. The figures and supplemented material is informative and clear as well.
Despite the small number of patients, the procedural approach has a wide range of options. I personally think that the paper is inspiring, and has merit.
Author Response
The authors present their experience with "percutaneous anorectoplasty" with low-infralevator anorectal malformations. Although the technique is described previously, this case series is presented with a wide range of percutaneous options in a relatively small number of patients which also represents a relatively rarer form of the malformation (as evidenced by only 10 patients in a 13 year period from multi-institutional collaboration).
The text is well written, clear and simple, and easy to follow. The figures and supplemented material is informative and clear as well.
Despite the small number of patients, the procedural approach has a wide range of options. I personally think that the paper is inspiring, and has merit.
> Thank you very much for the encouraging comments
Reviewer 3 Report
This article is a case series showing the short-term outcomes of patients who underwent PARP, which is the novel technique to treat selected patients with low-type anorectal malformations such as male perineal fistula or either sex without a fistula. I congratulate the authors on indicating excellent results of fecal continence as well as postoperative complications. However, there are some concerns, especially in terms of the patient selection and technical aspects and I recommend the authors address the following matters.
[Major]
I understand that in the PARP procedure, a surgeon punctures the distal end of the rectum and dilate the tract regardless of the usage of imaging modality. We usually detect the rectal pouch through the limited perineal skin incision maintaining the center of the muscle complex by using the electric muscle stimulator, and then, cut the distal end under direct vision. Are there any differences between yours and ours? What are the merits of your procedure? I mean why you bother to do such complicated manipulations.
The authors described that the rectal mucosa was retracted and anastomosed to the skin. However, I believe that all layers of the rectum should be pulled to the level of the skin after dissecting the rectum circumferentially and the neo anus should be created in a tension-free fashion. PARP technique seems to have a potential risk of postoperative wound dehiscence, which has a negative impact on functional prognosis as the authors described. Moreover, I think that it would be impossible to anastomose the rectum with the skin without separating the rectum from the surrounding tissue in many cases.
(p.7, line 218) In the discussion, the authors mentioned that the maximal distance between the rectal pouch and the skin was 3 cm. I am suspicious about the diagnosis of this patient. Was this case really the low-type ARM? The length of 3 cm is very long and I assume that this patient could be classified into the intermediate- or high-type ARM.
[Minor]
(p.2, line 62) I think “A Wangensteen-Rice radiograph” should be replaced with “invertogram” because the latter seems more common among pediatric surgeons.
(p.8, line 257) The authors discussed the possibility of double fistulas in their case of perioperative injury to the urethra. Does the case of double fistulas really exist? If so, please refer to related articles.
(p.8, line 275) What is the nPARP? The word was shown for the first time in the discussion without any previous explanations. Do you mean the PARP without image-guidance? Please indicate it.
Author Response
This article is a case series showing the short-term outcomes of patients who underwent PARP, which is the novel technique to treat selected patients with low-type anorectal malformations such as male perineal fistula or either sex without a fistula. I congratulate the authors on indicating excellent results of fecal continence as well as postoperative complications. However, there are some concerns, especially in terms of the patient selection and technical aspects and I recommend the authors address the following matters.
[Major]
3.1. Critique: I understand that in the PARP procedure, a surgeon punctures the distal end of the rectum and dilate the tract regardless of the usage of imaging modality. We usually detect the rectal pouch through the limited perineal skin incision maintaining the center of the muscle complex by using the electric muscle stimulator, and then, cut the distal end under direct vision. Are there any differences between yours and ours? What are the merits of your procedure? I mean why you bother to do such complicated manipulations.
> We feel that using ultrasound, radiography or endoscopy allows us to perforate through the center of the sphincter complex with a needle, limiting damage to the sphincter complex much like during the laparoscopic approach for higher lesions. Open dissection down to the rectal pouch may require more dissection and thereby cause more damage. This was added to the discussion in a paragraph on page 7.
3.2. Critique: The authors described that the rectal mucosa was retracted and anastomosed to the skin. However, I believe that all layers of the rectum should be pulled to the level of the skin after dissecting the rectum circumferentially and the neo anus should be created in a tension-free fashion. PARP technique seems to have a potential risk of postoperative wound dehiscence, which has a negative impact on functional prognosis as the authors described. Moreover, I think that it would be impossible to anastomose the rectum with the skin without separating the rectum from the surrounding tissue in many cases.
> We have added two sentences on this limitation to the discussion on page 7:
"Nevertheless, distance between the pouch and the skin may be a limitation of the PARP technique, making it applicable only to low-type lesions where the mucosa can be retract-ed downward and anastomosed to the skin. This approximation, however, results in a nicely inverted skin rosette and may prevent prolapse, which we have not seen as a com-plication in our series."
3.3. Critique: (p.7, line 218) In the discussion, the authors mentioned that the maximal distance between the rectal pouch and the skin was 3 cm. I am suspicious about the diagnosis of this patient. Was this case really the low-type ARM? The length of 3 cm is very long and I assume that this patient could be classified into the intermediate- or high-type ARM.
> Possibly so, but nevertheless we were able to retract the mucosa down to the skin, as described in the response to 3.2..
[Minor]
3.4. Critique: (p.2, line 62) I think “A Wangensteen-Rice radiograph” should be replaced with “invertogram” because the latter seems more common among pediatric surgeons.
> We have added the term "invertogram" in the text, as some may know it under that name.
3.5. Critique: (p.8, line 257) The authors discussed the possibility of double fistulas in their case of perioperative injury to the urethra. Does the case of double fistulas really exist? If so, please refer to related articles.
> According to the literature, the incidence of H-type anorectal malformation is around 3%. We have found 2 references which we have integrated in the revised manuscript, namely:
"According to the literature, such H-type anorectal malformations have an incidence of around three percent [25], ranging from 0.1 to 16 percent [26]. Therefore, pediatric surgeons should have a high index of suspicion when performing any of these procedures. Conversion to a PSARP in this our case 2 afforded the patient a good outcome"
3.6. Critique (p.8, line 275) What is the nPARP? The word was shown for the first time in the discussion without any previous explanations. Do you mean the PARP without image-guidance? Please indicate it.
> Thank you for pointing the lack of clarification. The nPARP is the procedure done without image guidance. We have clarified this in the methods on page 2 (paragraph 2.3.1.), as well as in the discussion on page 8, where the nPARP is discouraged.
- Other changes:
4.1. The Ethics registration number has changed to 22-0141. This change was updated.
Round 2
Reviewer 3 Report
I appreciate the authors for addressing my review comments sincerely. Thank you.